# Quantitative Synaptic Biology: A Perspective on Techniques, Numbers and Expectations

**DOI:** 10.3390/ijms21197298

**Published:** 2020-10-02

**Authors:** Sofiia Reshetniak, Rubén Fernández-Busnadiego, Marcus Müller, Silvio O. Rizzoli, Christian Tetzlaff

**Affiliations:** 1Institute for Neuro- and Sensory Physiology and Biostructural Imaging of Neurodegeneration (BIN) Center, University Medical Center Göttingen, 37073 Göttingen, Germany; sofiia.reshetniak@med.uni-goettingen.de; 2International Max Planck Research School for Molecular Biology, 37077 Göttingen, Germany; 3Cluster of Excellence “Multiscale Bioimaging: from Molecular Machines to Networks of Excitable Cells” (MBExC), University of Göttingen, 37077 Göttingen, Germany; 4Institute for Neuropathology, University Medical Center Göttingen, 37075 Göttingen, Germany; 5Institute for Theoretical Physics, University of Göttingen, 37077 Göttingen, Germany; 6Third Institute of Physics, University of Göttingen, 37077 Göttingen, Germany

**Keywords:** synapse, synaptic vesicle, quantification, modeling, imaging, super-resolution, cryo-tomography

## Abstract

Synapses play a central role for the processing of information in the brain and have been analyzed in countless biochemical, electrophysiological, imaging, and computational studies. The functionality and plasticity of synapses are nevertheless still difficult to predict, and conflicting hypotheses have been proposed for many synaptic processes. In this review, we argue that the cause of these problems is a lack of understanding of the spatiotemporal dynamics of key synaptic components. Fortunately, a number of emerging imaging approaches, going beyond super-resolution, should be able to provide required protein positions in space at different points in time. Mathematical models can then integrate the resulting information to allow the prediction of the spatiotemporal dynamics. We argue that these models, to deal with the complexity of synaptic processes, need to be designed in a sufficiently abstract way. Taken together, we suggest that a well-designed combination of imaging and modelling approaches will result in a far more complete understanding of synaptic function than currently possible.

## 1. Introduction

Synaptic efficacy and plasticity are key determinants of all brain functions, and of the corresponding behavioral output. Conversely, aberrant synapse transmission is the cause of many neurological and psychiatric disorders. It is therefore not surprising that synapses have been the focus of substantial numbers of quantitative analyses, providing information on virtually all aspects of their structure and function. Nevertheless, we are still unable to understand synaptic function on a sufficient level of detail to predict it with reasonable precision. This problem is deepened by the fact that individual processes, such as synaptic vesicle exocytosis or endocytosis, can be explained by a multitude of hypotheses, all of which are typically plausible, and therefore could, in principle, fulfill the respective functions. Such hypotheses led to arguments that still persist, decades after their inception (see for example [1,2,3]). How could this problem be solved? In this review we argue that the key problem is not that we lack an understanding of the biochemical functions of the different proteins. In fact, a remarkably high percentage of the proteins in the synapse have been thoroughly analyzed, more so than in many other cellular compartments. Synaptic structure has also been analyzed for more than 70 years, and, since synapses are more stereotypically organized than other cellular compartments, one could come to the conclusion that their organization should be perfectly understood. Nevertheless, novel organization concepts, as the segregation of parts of synapses in the form of liquid phases [4] appear regularly. Another important principle of synaptic organization is the formation of cholesterol-based domains that regulate the localization of a variety of proteins [5,6,7]. We suggest here that the main problem faced by synapse studies is that we do not understand the organization of synaptic proteins, in time and space, in sufficient detail. As discussed here, an emerging combination of wet lab tools and computational approaches should be able to overcome this difficulty. However, since the information that can be obtained in a wet lab will never be comprehensive, the choice of the appropriate modelling strategies becomes a fundamental aspect in understanding synaptic functions, possibly more important than the data-collecting strategies.

## 2. Quantitative Synaptic Analyses Provide Functional Insights and Indicate Potential Bottlenecks

Many types of quantitative synaptic analyses have been performed, from detailed electrophysiological investigations [8,9], to imaging studies in the optical [10,11,12] and electron microscopy domain [13,14], or complex biochemical studies of interacting synaptic proteins [15]. Several studies also attempted to integrate biochemical, electron microscopy, and imaging information to obtain detailed views of synaptic organelles (the synaptic vesicle [16] or entire synaptic sub-compartments [17], as indicated in Figure 1A,B). Crucially, such studies provided the copy numbers of the proteins involved, being within the context of the organelles or synapse areas. For example, the former study found that vesicle fusion proteins (SNAREs), calcium sensors, and neurotransmitter transporters were present in large copy numbers (10–70 per synaptic vesicle), and were thus unlikely to be rate-limiting in most functional steps, as exo- or endocytosis. In contrast, synaptic vesicles were found to contain, on average, one to two proton pumps, suggesting that this molecule can be a key bottleneck in vesicle recycling, since its loss would result in vesicles unable to release neurotransmitter. At the same time, the proton pump is functional when present on the plasma membrane, where it serves to alkalinize the cytosol [18]. Similarly, neurotransmitter transporters may also function when present on the plasma membrane [19], which implies that the regulation of their copy numbers in vesicles and on the plasma membrane is a critical element in the synapse. These assessments were largely confirmed in subsequent works [17,20], and have been used in numerous articles investigating, for example, vesicle acidification, exocytosis regulation, clathrin-mediated endocytosis, or synaptic vesicle dynamics in relation to neurodegeneration. Thus, quantitative studies support the identification of critical components in synaptic function. 

A subsequent analysis of the composition of the presynaptic bouton [17] provided copy numbers for multiple membrane and soluble proteins, which again have been used to investigate potential bottlenecks. It is generally agreed that synaptic vesicle exocytosis is followed by the retrieval of the vesicle proteins (endocytosis), and in turn followed by neurotransmitter refilling, which readies the vesicle for further rounds of exocytosis. It is still unclear how endocytosis is achieved, after more than 40 years of studies [1]. In principle, the fused vesicles can simply close their fusion pore, thus finishing the endocytosis process (kiss-and-run endocytosis [21]). Nevertheless, the involvement of a molecular cascade composed of clathrin and associated cofactors has also been shown to be important in endocytosis [22,23,24]. A further complication is that the clathrin involvement may take place not only at the plasma membrane, but also at a later step, on endosome-like organelles that initially formed without the involvement of clathrin at the plasma membrane [25,26]. In addition, the temperature at which experiments are performed may affect the involvement of clathrin [26,27]. Finally, endocytosis kinetics depend on the strength of the stimulus, with long, high-frequency stimulation trains resulting in much slower endocytosis than short stimulus trains (as already noted in the earliest studies [1]). 

This final aspect, the dependence of endocytosis kinetics on stimulus strength, could be explained by an analysis of the copy numbers of the cofactors in the clathrin pathway. Most such cofactors are present at about 2000–4000 copies for the average brain synapse (obtained from mixed rat cortex and cerebellum synaptosomes), an amount that is only sufficient for the recycling of a few vesicles at one time, since many copies of each cofactor are needed for recycling one vesicle (please note that these average synapses contained some 400 synaptic vesicles). For example, clathrin forms during endocytosis a spherical lattice that contains up to ~300 copies of the clathrin heavy and light chains [28], implying that one recycling vesicle would use about 10% of the clathrin complement of the synapse. Similar values were found for dynamin, which is involved in removing the vesicles from the membrane during endocytosis. As a consequence, the fusion of more than about 10 vesicles, occurring in vitro after stronger stimuli, will result in slow recycling, as the synapse runs out of cofactor molecules. Incidentally, this observation also offered an explanation for the relatively low number of synaptic vesicles that recycle in vivo at any one time; only a few percent of the vesicles can recycle, as the synapse is unable to endocytose larger fractions efficiently [29,30,31,32]. This example indicates that already the consideration of quantitative knowledge advances the understanding of synaptic processes. However, one should also take into account the fact that clathrin-independent endocytosis may take place (see Section 3 for more details).

Overall, this overview confirms our suggestion that copy numbers, as well as locations, are an important issue in synaptic analyses, as we discuss in further detail below.

## 3. Information of Protein Mobilities can Complement the Copy Number Information for More Detailed Functional Investigations

A major problem with static information on protein copy numbers is that the used methods have a low temporal resolution, or can even measure the state of the system at only one point in time. By contrast, the composition of the synapse is likely to change rapidly, as proteins diffuse across the limited synaptic space, especially as vesicles move and are exo- and endocytosed (see for example [34], and references therein). High motion rates have also been well established for the postsynaptic compartment, where several studies have analyzed neurotransmitter receptor movement, relying on tracking quantum dot-labeled receptors or on live photoactivated localization microscopy (PALM) of mEos-tagged receptors (see reviews [35,36]). Important components of the postsynaptic density (PSD), such as PSD95, have also been analyzed (see for example [37]) and a number of such studies have converged on the important hypothesis that the PSD forms because of a phase transition, with abundant PSD proteins such as SynGAP and PSD95 binding each other and forming a liquid-like droplet [38]. This droplet, in turn, determines the mobility of the PSD proteins and possibly also of the receptors through direct or indirect interactions like crowding effects [37,39]. Soluble proteins are, as expected, even more mobile, as observed for many signaling molecules that have been analyzed using FRET sensors constructed on their scaffolds (reviewed by [40]). As one is faced with a rapidly-moving synaptic environment, estimates on what may take place cannot rely on the average copy numbers of proteins from isolated boutons or dendritic spines, since protein recruitment to the synapse could play a fundamental role in any functional process. Protein recruitment, based on diffusion or on active transport, takes place on a time scale of seconds to minutes [33], and is therefore more relevant to rapid changes during activity than protein production, which may take minutes to hours.

As an example, the simple statement made above on how clathrin may eventually become limiting during prolonged stimulation may or may not hold true, depending on the mobility of this and related molecules. To showcase this effect, we relied on a recent study, in which the mobility of clathrin and of several other presynaptic proteins was estimated in cultured rat hippocampal neurons [33] (Figure 1C). Besides clathrin, endocytosis probably involves the GTPase dynamin [41,42] as well as the chaperone Hsc70 [43]. As the copy numbers of dynamin, clathrin, and Hsc70 needed for fission, coating, and uncoating of single vesicles [28,43,44,45], together with their overall population sizes in synapses [17,46], and their mobility have been estimated [33], we could then simulate their involvement in endocytosis relatively easily. The normal, “physiological” activity of the cultured neurons in which protein mobility has been recently analyzed [33] consists of short bursts of action potentials, delivered at a burst frequency of ~0.1 Hz and releasing ~six synaptic vesicles per burst [47]. We simulated this situation (Figure 2A) by considering the release of six vesicles at every 10 s. Afterwards, dynamin alone (top path), or both dynamin and clathrin (bottom path), needed to accumulate on the fused vesicles in sufficient copy numbers, before endocytosis could take place. Hsc70 was then collected on the coated vesicles, to uncoat them, and to allow clathrin molecules to return to the soluble pool, and to participate in further endocytosis steps (Figure 2A). Using the mobility data discussed above, we calculated the accumulation of the proteins on single fused vesicles with a time resolution of 1 s. When sufficient molecules accumulated, the vesicles were considered to progress to the next step. For example, a vesicle could be considered as having been endocytosed only after the accumulation of sufficient copies of dynamin (if dynamin alone was considered in the respective simulation), or of dynamin and clathrin. It then became a coated vesicle, which could only be considered to be uncoated after the accumulation of sufficient Hsc70 copies. Before uncoating, the clathrin molecules were considered to be bound to the particular vesicle and were therefore unable to participate in the endocytosis of other vesicles. 

The results indicated that the amounts of dynamin should not be limiting under these conditions (Figure 2B), implying that dynamin could be involved in every endocytosis step (as has been demonstrated in *C. elegans* [24]). However, an absolute requirement for clathrin in endocytosis cannot be supported under these spontaneous network activity conditions, since exo- and endocytosis cannot be balanced (Figure 2B). Even if the quantity of clathrin needed per vesicle is assumed to be as low as seven triskelia, endocytosis cannot balance exocytosis for more than a few bursts (Figure 2D). Exo- and endocytosis are balanced reasonably well if clathrin is needed for only one out of each six vesicles released during a burst (Figure 2C), but not when more vesicles require it. This suggests, in line with several recent observations [22,23,24,48], that the formation of the clathrin coat is probably not required for every vesicle endocytosis event, even under mild physiological activity. In principle, one could also envision more complex endocytosis mechanisms, in which the synaptic vesicles bring along some of the necessary clathrin molecules. Such mechanisms did not fare well in our simulations, since having clathrin bound to vesicles removes it from the mobile clathrin pool and actually damages the chances of the vesicles being endocytosed (data not shown). Other hypotheses could envision high concentrations of clathrin at the active zone or peri-active zone. While this may allow a stronger clathrin involvement during the first round of endocytosis, the diffusion of clathrin during uncoating is strong enough to distribute the molecules in a wide area, so that the next rounds of exo- and endocytosis would not benefit from high clathrin concentrations at the activity sites. Importantly, assuming substantially higher speeds or higher available molecule pools for the clathrin and Hsc70 molecules changes this result, enabling clathrin to participate in the endocytosis of every synaptic vesicle. This makes it evident that any assumption on how clathrin is involved in synaptic transmission needs to take into account realistic synapse conditions. At the same time, this example suggests that clathrin can be involved in a substantial number of recycling events. It is probable that clathrin and its cofactors are especially required when vesicle molecules disperse in the plasma membrane, and need to be re-sorted into functional vesicles [2,3,49]. This event may not take place too frequently during mild physiological activity [3], during which the vesicle may persist as meta-stable assemblies that are recycled as a whole (see further discussion in [47,50,51]). Such events may take place during supra-physiological stimulation, implying that clathrin may become limiting under such conditions.

Taken together, the discussed example shows that the consideration of quantitative knowledge of copy numbers and of mobility data of proteins, as well as the usage of mathematical models to integrate such data from different methods, considerably advances the understanding of the dynamics of synaptic processes. 

## 4. The Main Dangers: Missing Information and False Information

Much of how we understand synaptic function comes from our perception of the synapse organization, and of the arrangements of its proteins. However, in spite of almost 15 years of synaptic investigations by super-resolution microscopy [10,52], substantial information is missing on the location of synaptic proteins. Furthermore, there are several sources of error in imaging approaches; some of which also apply to other methods.

To showcase this, in the following we turn again to synaptic vesicle proteins, as an example. Synaptic exocytosis needs to take place with sub-millisecond kinetics, for accurate neurotransmission. These kinetics are difficult to understand, because the SNARE proteins responsible for exocytosis “intrinsically operate on a timescale of about a second” [53]. The solution can only come from a special arrangement (organization) of SNAREs and related proteins. For example, the Ca^2+^ sensor synaptotagmin-1 has been proposed to form a ring-like arrangement at the interface between docked synaptic vesicles and the plasma membrane, which helps in modulating the exocytosis kinetics [53]. Other vesicle proteins may also be arranged in clusters or ring-like arrangements that may aid in exocytosis. These include the SNARE synaptobrevin2/VAMP2, or synaptophysin, a vesicle marker with four transmembrane domains [54]. Vesicle protein clusters have also been proposed to aid in endocytosis [1,2,3]. The clusters are presumably targeted and retrieved by the endocytosis machinery, and, if the vesicle proteins were unable to form clusters, they would eventually scatter on the plasma membrane after exocytosis, and would be much more difficult to sort, retrieve, and endocytose. Albeit attractive, this idea of the involvement of vesicle protein clusters in exo- and endocytosis has one major fallacy; sub-vesicular protein clusters have never been seen in real vesicles, with only some hints obtained so far from electron cryo-tomography [54,55,56]. The main reason is the insufficient imaging resolution (Figure 3), since commonly-used super-resolution approaches are simply unable to differentiate between these hypotheses. 

### 4.1. Sample Preparation and Fixation

Low imaging resolution is not the only difficulty encountered here. Sample preparation and the identification of the molecules using different types of probes has long been a concern, and has only become more so over the last decade, as the improving microscopy precision implies that the tolerance for artifacts is increasingly lower. The large majority of microscopy studies rely on chemically fixed samples, in which particular proteins are then revealed by the use of immunolabeling probes (typically antibodies). The typical protocol involves fixation using an aldehyde solution, with the 4% paraformaldehyde (PFA) being the preferred condition, for periods ranging from 10 to 60 min. The sample is then permeabilized using a detergent (typically Triton ×100, at 0.1–0.5%), to enable the penetration of antibodies. The sample is then incubated with primary antibodies, to reveal the protein of interest, and then with secondary antibodies, to reveal the primary ones. Every one of these steps suffers from substantial difficulties. First, PFA fixation results in changes to the morphology of the samples and it induces the mislocalization of many target proteins, in an unpredictable fashion, due to the fact that it penetrates into the samples with difficulty and fixes them slowly (for example [58,59,60]). The samples can remain alive for minutes after the application of PFA [61] and continue to engage in various biological functions, such as exo- or endocytosis [62]. As some cellular components slowly become fixed, these functions are increasingly disturbed, implying that the final overview of the fixed cell may be substantially different from that of the live cell. Another major issue is that even exhaustive PFA incubations are unable to fix the samples thoroughly. In general, about half of the proteins are fixed [62], with the effect particularly bad for cytosolic proteins, which are then washed off during the permeabilization step [62]. In principle, these problems could be alleviated by combining PFA with a faster- and stronger-coupling aldehyde, as glutaraldehyde [59]. However, such mixtures tend to reduce antigenicity [63], while not solving the problem of insufficient cytosol protein fixation, which is immediately observed when comparing PFA/glutaraldehyde fixations with pure glutaraldehyde fixations in conventional electron microscopy [64]. Another solution is the use of glyoxal, a di-aldehyde that acts as an intermediate step between the slow-acting PFA and the aggressive glutaraldehyde [62]. Overall, the fixation step needs to be calibrated very carefully, preferably by comparing the protein of interest in live and fixed cells.

### 4.2. Permeabilization of Cell Membranes

Following fixation, a permeabilization step is required for allowing epitope recognition by the antibodies. Various detergents are used to disrupt the cell membranes, removing lipids from the membranes, and causing hydrophobic patches to be exposed. This results in membrane collapse on the nanoscale and may be an important cause of membrane patterning, as no continuous labeling can be obtained in a discontinuous membrane. Fortunately, experiments with non-permeabilized membranes also demonstrated the nano-patterning of large numbers of proteins (for example [7]), but this concern still remains for every new target. Importantly, synaptic proteins interact with lipids in many phases of synaptic activity (for example [65]), and form specialized domains, whose identity and stability depend on the presence of specific lipids, including cholesterol or phosphoinositides, both on the presynaptic and the postsynaptic sides [7,66,67].

### 4.3. Antibodies

The subsequent antibody application may be the least precise step in the entire procedure. IgGs are among the most commonly employed tools for protein imaging [68,69], but their quality and reproducibility are known to be inconsistent [70,71], even for conventional imaging assays. At super-resolution, several problems become apparent, caused by the divalent structure of the antibodies. Not only the primary antibodies, but also the secondary ones attempt to bind two targets. Combined with insufficient fixation (as discussed above), this implies that the primary antibody may bring two target proteins together, while the secondary may collect two primary antibodies. As polyclonal antibodies are generally used as secondary reagents, multiple secondaries may bind one primary antibody, increasing the antibody-collecting possibilities and resulting in substantial protein clusters (Figure 4; see [72,73,74]). Such clusters were especially prominent in the first years of super-resolution imaging, to the point of becoming a strong concern (see review [75]).

Current research still employs antibodies for high-resolution imaging, but smaller, monovalent probes are replacing them slowly. Two prominent categories are the aptamers and nanobodies, which we explain here in brief. Aptamers are small DNA or RNA oligonucleotides [76], which are selected in vitro by a process termed ‘systematic evolution of ligands by exponential enrichment’ (SELEX), in which a library of single-stranded DNA or RNA, consisting of about 10^15^ sequences, is incubated with the protein of interest. The few sequences that can bind the protein of interest are retained, while all others are washed off. This procedure is then repeated several times, under increasingly harsher binding conditions, until sequences of suitably high affinity are selected. The resulting aptamers are directly coupled to the desired fluorescent dyes and are used in microscopy studies, where, being much smaller than antibodies (~2–4 nm, as opposed to more than 10 nm for antibodies), they penetrate better into the tissue and identify with more ease the epitopes [73]. The same observation was made with small camelid antibody fragments, nanobodies [72]. Nanobodies, which are similar in size to the aptamers, are produced by injecting the protein of interest into camelids (typically llamas). The animals’ B-cells produce both conventional dual-chain antibodies and single-chain antibodies for the protein of interest. The sequences of all single-chain antibodies are subsequently derived from a cDNA library obtained from the animal’s B-cells. The precise sequence of the single-chain antibody that binds the protein of interest (the nanobody) is then obtained by a phage-display procedure, is then produced in bacteria and is directly linked to fluorescent dyes for microscopy experiments [77]. The use of these tools should solve some of the problems induced by the antibodies, including the need to rely on permeabilization, since the small probes can enter fixed cells without the need of detergent treatments [78]. The main remaining problem is that such probes are only available for a handful of synaptic targets, albeit a compromise solution would be to use nanobodies as secondary reagents, which alleviates much of the clustering induced by secondary antibodies [79,80].

### 4.4. Imaging Resolution

These improvements in protein labeling are currently being mirrored by improvements in imaging resolution. It is generally known that imaging resolution has been limited by diffraction to about half of the wavelength of the imaging light (in most microscopes, about, 200 nm to 300 nm). Two main strategies have been generated to overcome this issue. First, the so-called coordinate-targeted approach, in which a beam of light is patterned and is applied onto the specimen in a fashion that enables reading the coordinates from which specific fluorophores are allowed to emit. Examples for this approach come from the stimulated emission depletion microscopy group of technologies (STED; [81]) and from the saturated structured illumination microscopy group (SIM; [82]). SIM currently reaches 60 to 100 nm in resolution. The majority of STED experiments with biological samples reach between 40 and 50 nm in resolution (for example [17]). Second, the single-molecule approach, in which the positions of single fluorophores that emit randomly are determined. This approach is typical for photo-activated localization microscopy (PALM; [83]), stochastic optical reconstruction microscopy (STORM and dSTORM; [84,85]), or ground state depletion microscopy followed by individual molecule return (GSDIM; [86]). These techniques typically reach 20 to 30 nm in biological experiments. Even higher resolution can be obtained using the MINFLUX approach (maximally informative luminescence excitation; [87]), which unifies a coordinate-targeted approach (as in STED microscopy) with the precise localization of single fluorophores (as in STORM microscopy). Molecules are turned on stochastically, as in STORM, which implies that molecules that can “blink” efficiently, such as Alexa647, are necessary for this technique. The molecules are then interrogated using an excitation beam that is patterned in the shape of a doughnut, with an intensity minimum in the center. This beam is scanned rapidly across the space containing the fluorophores, and elicits fluorescence only when its “edge” encounters a fluorophore. As the beam center moves over the fluorophore, fluorescence is no longer emitted, since the beam center has a low intensity. Since the position and the shape of the beam are known with high precision at all times, the position of the fluorophore can be determined rapidly, after only detecting a handful of photons. This results in a measurement resolution of 1–4 nm in biological samples [87,88], even in multiple color channels [89], and has recently entered the synapse domain (Figure 5A–C).

Importantly, major advances in cryo-electron tomography (cryo-ET) methods have recently enabled the description of particular molecular assemblies in cells, such as proteasomes or polyglutamine inclusions [92,93]. Although this technology is still difficult to apply to a general analysis of protein organization in cells, the combination of cryo-ET with direct electron detectors, phase plates and cryo-focused ion beam milling (cryo-FIB) has opened up immense possibilities for studying the molecular organization of synapses pristinely preserved by vitrification. A few studies have already been undertaken to image presynaptic architecture [91,94], revealing that a dense network of filaments link synaptic vesicles to each other (“connectors”) and to the active zone (“tethers”; Figure 5D,E), and recent progress in identifying molecular complexes promises to rapidly drive this technology further (Figure 5F). This work is built on previous studies performed using quick freezing, followed by freeze-fracture, etching, and finally rotary angle shadowing [95,96]. Overall, cryo-ET hopes to solve the problems indicated above. It preserves the samples optimally, in a life-like status. It does not require permeabilization or antibody labeling, and it has a resolution beyond any fluorescence approaches. Therefore, further progress in cryo-ET technology will be eagerly awaited by a large biological community.

Thus, experimental data and insights are continuously changed by new technological developments, but also by diverse sources of errors; although recent technology advances offer exciting potential solutions.

## 5. Are Differences between Synapses in Different Regions and Organisms too Great to Enable Parameter Transfer between Models? 

The perspective we argue for, that a quantitative analysis of the synapse would result in data that would enable a better understanding of synaptic function, derived through synaptic modelling, is only valid as far as data from different studies can be combined. While in general lines synapses seem to employ the same protein machineries, presumably for similar functions [2,3,49], it is difficult to demonstrate that quantitative parameters are similar in different types of synapses. The overall arrangements of particular molecules may appear similar, when investigated using super-resolution imaging in small hippocampal synapses in culture, or in large neuromuscular synapses [17], but this does not confirm that functional parameters are also similar. 

Several reviews have analyzed the quantitative parameters of different synapses in different organisms (for example [24,97]). In brief, some of the best-quantified synapses are neuromuscular junctions from the frog, *Drosophila,* and *C. elegans*, along with small synaptic boutons from the rodent hippocampus, the large calyx of Held synapse from the auditory pathway, and the highly-specialized lamprey reticulospinal giant synapses. These synapses contain widely different levels of synaptic vesicles, from ~200 vesicles in the hippocampal synapses and in the *C. elegans* boutons, to thousands of vesicles in the lamprey, tens of thousands in *Drosophila*, and hundreds of thousands at the frog neuromuscular junction and in the calyx of Held (see [24,97] and references therein). The pools of vesicles immediately available for release also vary over three orders of magnitude among these synapses. It is therefore unclear whether the same mechanisms would function at these synapses, with quantitatively similar parameters. Some indications are that this may be the case. For example, vesicle pool dynamics appear similar [97], and endocytosis mechanisms are remarkably similar (compare references [25,26] and [98]). Nevertheless, since quantitative, multi-protein synaptic studies are rare, comparisons across multiple types of synapses have not yet been thoroughly made. It is still unclear whether comparative levels of different synaptic proteins can be found in these synapses (see [46] for a method that may enable such quantifications).

However, a recent observation suggests that synapse heterogeneity may not be as much of a problem as one may envision. Two papers studied the dynamic organization of proteins in rat hippocampal synapses, in culture and in mouse neuromuscular synapses, at an interval of almost a decade [30,33]. The first study investigated the strength of the association between soluble synaptic proteins and synaptic vesicles by perturbing the synaptic vesicles using black widow spider venom, which, when applied in the absence of Ca^2+^, causes massive exocytosis in the absence of endocytosis, thereby depleting the vesicle clusters [99]. Soluble proteins binding the vesicle cluster diffused out into the axon (Figure 6A), which enabled an estimation of the buffering of proteins by the synaptic vesicles. The more recent study expressed proteins in hippocampal neurons in culture and employed fluorescence recovery after photobleaching (FRAP) to obtain estimates of their movement (Figure 6B,C). The movement, both within the synapses and in the vesicle cluster, was estimated, and the diffusion coefficients of proteins among the vesicles were determined. Importantly, a strong correlation between the estimates obtained in the mouse neuromuscular synapses and in the hippocampal cultures was obtained (Figure 6D). 

Overall, combining the two studies demonstrates that proteins that are more strongly buffered by vesicles also move more slowly in the vesicle cluster. This is not surprising, but the level to which the two sets of estimates correlate is remarkable, since they were obtained in widely different preparations, and with widely different technologies. This suggests that quantitative parameters obtained in different types of synapses may be combined in order to obtain more insight in modeling studies.

## 6. The Basis of Mathematical or Computational Models to Understand Synaptic Function

Mathematical or computational models are widely used in different scientific fields. All of these have in common that they rely on a certain level of abstraction. This implies that the benefit of models is not to capture all details of the subject to “mimic the real world”, but to extract general principles, which determine the dynamics of the system. Such a general principle could describe, for instance, the temporal course of a process, or be a linear function describing the relation between two processes. Variability between synapses could support the derivation of a general principle, as each synapse could provide a new data point of the relation between the processes. Different types of synapses, e.g., the calyx of Held compared to a synapse between pyramidal neurons, could be represented by different parameter values of a general principle (e.g., a different slope of the linear function). In general, computational models of a synapse will be far more complex, formalizing a large number of processes and their relations (see e.g., [100] for a model describing about 1100 chemical reactions of the signaling pathways controlling long-term plasticity) that could depend on time and space.

Of course, each level of abstraction implies different advantages and disadvantages. In the context of understanding synaptic function, very detailed computational models describe synaptic processes and their relations on the level of single molecules or even atoms. Such models are mainly considered in molecular biology to investigate, for instance, membrane dynamics [101] or the functioning of single channels [102]. The molecular simulations are able to provide valuable insights into judiciously selected processes, such as those that alter the membrane topology in exo- and endocytosis [103,104]. In particular, molecular simulations often have a higher spatiotemporal resolution than experiments, can simultaneously access qualitatively different properties (e.g., molecular conformations of lipids and proteins, local membrane shape, and free-energy differences), and independently vary parameters (such as membrane composition and tension, or the presence of proteins) that may be strongly coupled in an experimental system. Thus, detailed computational models provide a rather unbiased view of the molecular processes on short time and length scales. In addition, they enable the qualitative correlation of different determinants that underlie synaptic processes on the molecular scale, e.g., information of the mechanisms of fusion and fission, the role of membrane composition or local curvature [105], as well as the role of fusion [106] and fission proteins [107,108]. 

The choice of the level of abstraction for molecular models, as well as for all theoretical models, depends on the scientific question, and is dictated by a balance between more detailed descriptions and required analyses and computational resources. Detailed molecular models build on sophisticated and carefully developed atomistic force fields, and thereby invoke rather few assumptions. By contrast, more coarse-grained models allow access to larger scales (several tens of nanometers and microseconds) and systematic parameter studies, while field-theoretic models that only capture the universal features of lipid architecture, allow straightforward access to free energies [109,110]. Therefore, current molecular models range from chemically realistic representations, like the CHARMM force field [111], over the coarse-grained MARTINI model [112,113], to top-down, implicit-solvent model [114] and meshless membrane representation [115]. For instance, the analysis of selected aspects of clathrin-dependent and -independent vesicle endocytosis depicted in Figure 2A challenges the scales accessible to detailed molecular modelling, and thus requires coarse-grained models. Whereas early coarse-grained models chiefly focused on representing the molecular structure, more modern ones, e.g., the MARTINI model [112], also include thermodynamic information and have been developed for an ever increasing selection of molecular compounds. One of the big challenges consists in an accurate representation of dynamic processes [116,117]. Different coarse-grained models focus on accurately representing different thermodynamic, structural, or dynamic properties. Thus uncertainties due to the underlying coarse-grained model depend on the specific property. In general, coarse-grained models rather accurately represent large-scale equilibrium properties, such as the compression and bending moduli of membranes or membrane-mediated interactions, because these properties are related to universal aspects of self-assembly in amphiphilic molecules. Note that the excess free energy of highly bent structures, such as fusion stalks or hemifission structures, is more sensitive to microscopic model details, as they strongly depend on, e.g., the molecular architecture and interactions. Nevertheless, coarse-grained models are expected to capture semi-quantitative correlations, such as the dependence of the excess free energy of stalks on hydration. If experimental results are not available, an alternative for validating results consists in comparing the predictions of different coarse-grained models. 

Much of the recent success of coarse-grained models stems from the increasing computational power, model developments, e.g., extension of coarse-grained force fields to describe proteins [118], and advanced simulation [119,120] and free-energy techniques [110,121,122]. The latter techniques help to identify pathways and concomitant free-energy barriers, thereby significantly expanding the range of time scales that are accessible to molecular simulation. Such models have established the role of membrane tension and lipid composition [109], and the role of hydration repulsion between membranes [106 and membrane curvature [105], suggesting additional control strategies. Whereas pore formation and fusion have attracted abiding interest, related phenomena such as fission or spreading of vesicles on substrates remain comparatively less explored.

A detailed molecular model of the whole synapse could be used to investigate the molecular underpinnings of certain diseases and the mode of action of medicines. However, atomistically detailed modelling of all processes in an entire synapse, and even coarse-grained particle-based models, will remain computationally unfeasible in the next decade(s); the time and length scales (minutes to hours and several micrometers) are simply too large for direct particle-based simulations. Furthermore, due to its detailedness, the susceptibility of such a model parameterization to experimental inaccuracies or variations, as discussed above, needs to be managed carefully.

Alternatively, abstract computational models summarize the essential dynamics of the synapse in rather simple mathematical formulations using coupled differential equations [123,124]. Such abstract models are a standard approach in network neuroscience, and they enable scientists to link synaptic dynamics to cognitive mechanisms (which are mainly associated with network dynamics) such as learning, memory, or decision making, operating on time scales from milliseconds up to several days. Nevertheless, such an abstract model of the whole synapse would strongly simplify the complex dynamics within the synapse making the link to quantitative data of synaptic proteins difficult.

Although there is a gap between molecular models and the more abstract computational models employed in network neuroscience in terms of time and length scales (and therefore in terms of investigated phenomena), there are also multiple points of contact. On the one hand, the qualitative correlations between determinants that underlie synaptic processes on the molecular scale may suggest relevant degrees of freedom in more abstract neuroscience models, and inform functional dependencies employed in these approaches. On the other hand, the abstract neuroscience models, in turn, provide information about the spatiotemporal changes of the local variables at the site of fusion and fission in the course of synaptic function, such as concentration of proteins or signaling molecules. This information can set the boundary conditions for coarse-grained molecular models and thereby focus the exploration of parameters on relevant regions. 

Of course, the above-discussed examples describe two extremes of possible abstraction level to describe synaptic functioning. Between the discussed molecular and network level is a wide range of possible levels of abstraction, some being covered by computational models. On each level of abstraction, to derive the general principles via carefully designed, parameterized and devised computational models, quantitative data from different methods is required. For instance, data of polymerization and depolymerization of actin filaments, imaging data of the spatial distribution of actin in the post-synapse, and a model of membrane dynamics have been integrated to investigate the spatial-temporal dynamics of a dendritic spine [125]. This model enabled the systematic analyses of the impact of ongoing actin dynamics and their lifetime on the spine size and shape. Furthermore, the computational model facilitated the derivation of a simple mathematical relation between the number of polymerization foci of actin and the spine size. This relation constitutes a potential, previously unknown general principle of synaptic function, which would have been difficult to derive only based on experimental data. By contrast, this new general principle yields new predictions, possibly stimulating new experiments advancing the understanding of synaptic function.

Thus, to advance the understanding of synaptic function, we think that a set of models across different abstraction levels is required. For this, the synapse has to be subdivided into loosely dependent processes. This distinction can be made based on physical space, known function, involved proteins, etc. The functioning of these individual processes, happening on a rather detailed level, can already be investigated using quantitative experiments and molecular models. On a higher level of abstraction, several of these processes can be combined and integrated into more abstract computational models. This allows the investigation of their interplay and the validation of the results by additional experiments. These abstract computational models of different processes can again be combined and integrated on the next level of abstraction. For instance, the presynaptic processes of vesicle supply, inactivation, and recovery have been integrated into the Tsodyks and Markram model of short-term synaptic plasticity [126,127], while the calcium and AMPAR dynamics in the post-synapse have been combined in models of spike-timing-dependent plasticity [128,129]. Now, by linking both types of model, the interplay between all of these processes and, thus, of pre- and postsynaptic function has been investigated and, for instance, can be related to memory function, providing a more detailed understanding of synaptic plasticity [130,131]. We are confident that a similar “hierarchical” approach, being across several levels of abstraction, and integrating experimental and computational efforts, would also advance the investigation and comprehension of synaptic functioning.

## 7. Conclusions

Overall, we argue here that future research will benefit from sub-dividing the synaptic space and the synaptic functionality in compartments that can be investigated with molecular precision, independently of one another. Such processes may include movement and organization in the vesicle clusters, or active zone organization and function during stimulation, on the presynaptic side, as well as receptor movement and clustering, or PSD dynamics on the postsynaptic side. As long as sufficiently precise data can be obtained, covering multiple proteins in each compartment, this would enable computational models that allow us to predict the spatiotemporal dynamics of each compartment. These could then be integrated into larger-scale models, ultimately resulting in reasonably precise models of synaptic function, which would be able to offer new hypotheses for experimental testing, in synaptic function and dysfunction. While some gaps are still apparent, especially concerning the ability to image synapses at molecular resolution, or the understanding of synaptic protein kinetics, these are increasingly covered by the advancing research in the fields of both fluorescence and electron microscopy, thus offering the hope that functional synaptic models will become available in the next decade.

## Figures and Tables

**Figure 1 ijms-21-07298-f001:**
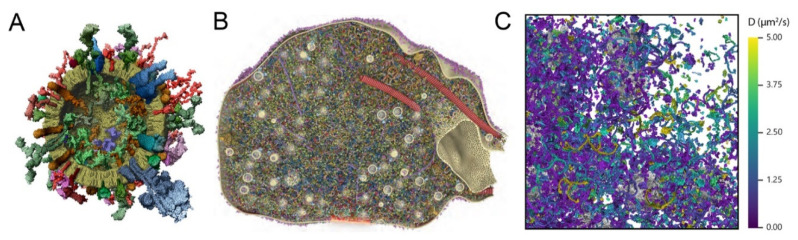
Quantitative synaptic views. (**A**), Molecular view of a synaptic vesicle, in which the different molecules are placed randomly, in copy numbers corresponding to quantitative biochemistry measurements. The large blue molecular complex to the lower right is the proton pump [16]. (**B**), A reproduction of the presynaptic bouton, indicating 60 different proteins, shown at copy numbers determined from quantitative biochemistry on mixed rat cortex and cerebellum synaptosomes, and in positions corresponding to an analysis performed by super-resolution imaging [17]. (**C**), A dynamic view of the presynaptic space. Proteins are shown in their original shapes (same as in (**B**)), but color-coded according to their calculated diffusion coefficients. Modified from [33]. For all panels, the synaptic vesicles provide a size scale, as they are 42 nm in diameter (clearly visible in panels (**A**,**B**); shown in gray in panel (**C**)).

**Figure 2 ijms-21-07298-f002:**
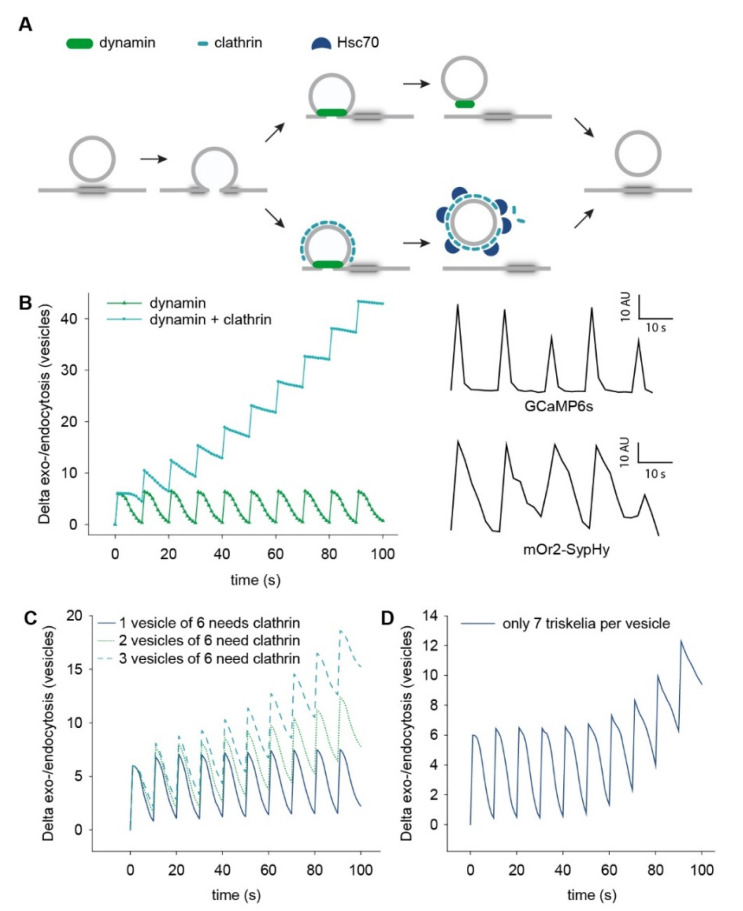
Simulations of the usage of three endocytosis cofactors during spontaneous network activity, (**A**), models of endocytosis tested in the simulations. To be endocytosed, each vesicle either needs to accumulate a sufficient number of dynamin molecules, which are quickly released after endocytosis, or, in addition to dynamin, a full clathrin coat is also required, which is then removed by Hsc70. (**B**), Results of the simulations showing the difference between the number of exo- and endocytosed vesicles in both of the tested hypotheses. Dynamin is supplied in necessary copy numbers, and can be involved in endocytosis at every step, while in the case of an absolute requirement for clathrin, endocytosis does not balance exocytosis. The inset to the right shows an indication of the spontaneous network activity modeled here, in the form of fluorescence intensity changes of a calcium sensor (GCaMP6s, top), and of a vesicle endo-/exocytosis indicator (mOr2-SypHy, bottom) recorded in live boutons during spontaneous activity. For the full statistics of such measurements, see [47]. (**C**), We tested whether endocytosis can balance exocytosis when only a fraction of the released vesicles require a full clathrin coat. When one vesicle out of six required clathrin, exo- and endocytosis are balanced reasonably well. (**D**), Exo- and endocytosis can also not be balanced when as few as seven triskelia are required for each vesicle. The models were performed based on data obtained in cultured rat hippocampal neurons, as indicated in the main text.

**Figure 3 ijms-21-07298-f003:**
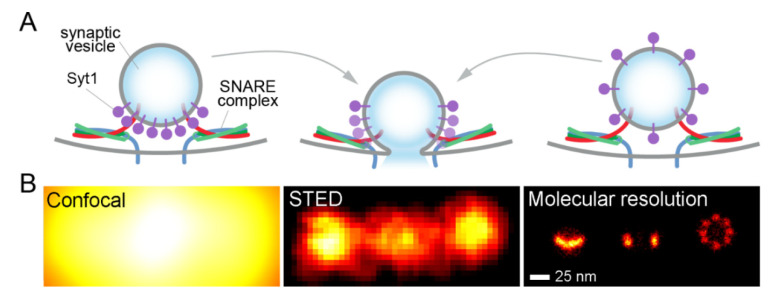
A fundamental difference between super-resolution and molecular-scale resolution. (**A**), A cartoon description of synaptic exocytosis. This process is driven by SNARE proteins under the control of the Ca^2+^ sensor synaptotagmin-1 (Syt1, purple). Syt1 has been suggested to form a ring next to the plasma membrane (left), which prevents fusion in the absence of Ca^2+^ [57]. Alternatively, it may be distributed along the vesicle membrane (right). (**B**), We modeled the 2D Syt1 images in (**A**) using realistic parameters for confocal (left), STED (middle) or molecular-scale resolution (right). The different hypotheses can only be tested by molecular-scale resolution.

**Figure 4 ijms-21-07298-f004:**
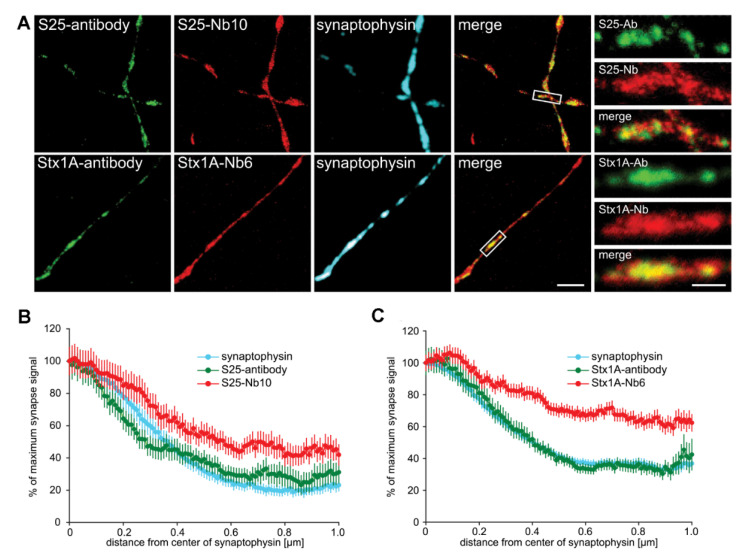
Typical antibody-induced artefacts in synaptic imaging. (**A**), The exocytosis SNARE proteins SNAP25 and syntaxin1A were imaged using stimulated emission depletion microscopy (STED) microscopy in cultured hippocampal neurons, relying on antibodies (green) or nanobodies (red). Synapses are indicated by a conventional antibody immunostaining for the synaptic vesicle marker synaptophysin (blue). Both the overview images and the insets give the impression that the antibodies form large spots or clusters, while the nanobodies provide a smoother pattern. Scale bars: 2 µm (left panels) and 0.5 µm (right panels). (**B**,**C**) An analysis of the signal patterning, in terms of signal loss in relation to the center of the synapses revealed that the antibodies indeed formed large clusters, mostly in synapses, while the nanobodies also detected the molecules elsewhere. Reproduced from [74].

**Figure 5 ijms-21-07298-f005:**
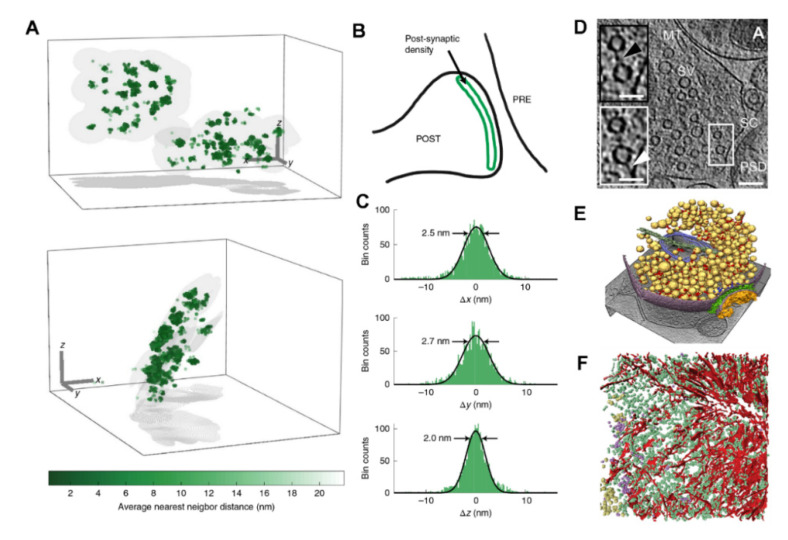
Study of neuronal architecture by maximally informative luminescence excitation (MINFLUX) and cryo-electron tomography (cryo-ET). (**A**), MINFLUX images of the post-synaptic scaffold protein PSD95. The color maps indicate the 3D density of the molecules. The gray areas are curved surfaces on which the molecules lie. The molecules were directly bound to a chemical fluorophore. (**B**), Theoretical organization of the molecule. (**C**), Histograms of localization precision for the molecules shown in panel A, indicating an overall precision of 2.0–2.7 nm. Reproduced from [89]. (**D**) Tomographic slice showing presynaptic short filaments that link synaptic vesicles to each other (“connectors”; black arrowhead in top inset) or to the active zone (“tethers”; white arrowhead in bottom inset). The insets show the same vesicles on different tomographic slices. MT: microtubule, PSD: postsynaptic density, SC: synaptic cleft, SV: synaptic vesicle. Scale bars: 100 nm main panel, 50 nm insets. (**E**) 3D segmentation of a presynaptic terminal showing synaptic vesicles (gold), the presynaptic membrane (purple), densities present in the synaptic cleft (light green), the postsynaptic density (orange), a microtubule (dark green), and a mitochondrion (light blue). Connectors (red) and tethers (blue) were detected automatically [90]. (**D**,**E**) are reproduced from [91]. (**F**) 3D segmentation of a neuronal c9orf72 poly-GA aggregate (red) and different macromolecules including ribosomes (yellow), TRiC (purple), and proteasomes (green). Reproduced from [92].

**Figure 6 ijms-21-07298-f006:**
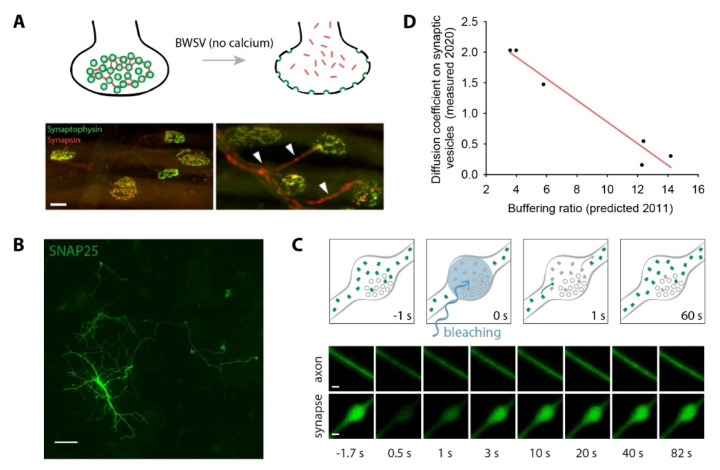
Estimates of protein mobility in synapses, from different technologies. (**A**), Synaptic vesicle clusters were disrupted by application of black widow spider venom (BWSV) on freshly dissected neuromuscular synapses. As the vesicles (green, marked by synaptophysin) are lost by exocytosis not compensated by endocytosis, the soluble proteins diffuse out of the boutons (red; synapsin is shown in the example, showcased by the arrowheads in the right panel, corresponding to the BWSV application). Scale bar: 20 µm. Reproduced from [30]. (**B**), Synaptic proteins were expressed in cultured hippocampal neurons (green). Scale bar: 100 µm. (**C**), This was followed by a fluorescence recovery after photobleaching (FRAP) procedure, in which synapses or axonal areas were bleached, and the movement of proteins within the bleached areas was estimated. Scale bar: 500 nm. (**B**,**C**) reproduced from [33]. (**D**), Both technologies produced estimates of protein binding to vesicles, either as protein buffering ratios (X axis, from neuromuscular synapses), or as diffusion coefficients in the vesicle cluster (Y axis, from hippocampal synapses). The two measurements correlate well for the six common proteins investigated (R^2^ 0.95, *p* < 0.001).

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
