# Peer review of "Quantitative Synaptic Biology: A Perspective on Techniques, Numbers and Expectations"

_ijms, 2020, doi:10.3390/ijms21197298_

Round 1
Reviewer 1 Report
In this review paper, Reshetniak and colleagues provide an insightful and critical overview of the knowledge and challenges in the quantitative description of synaptic biology. In recent decades, an impressive effort from many labs aimed to identify and quantify the key molecular players at both pre- and post- synapse. Although the repertoire of molecules is well-established, the key concepts that underlie synaptic organization and function are still elusive. In this review, the authors provide a critical overview of the challenges in the field, particularly focusing on:
(i) What the rate-limiting steps are in synaptic processes (i.e., acidification of SVs, clathrin-coat assembly/disassembly, receptor delivery to the postsynapse).
(ii) Technical challenges that arise during data acquisition and interpretation (e.g., variability due to sample preparation, limits of microscopy techniques, labeling, processing extensive data).
(iii) The mobility of proteins sequestered by SVs as a general feature conserved among different types of synapses, further suggesting a common mechanism behind this dynamic sequestering/buffering.
(iv) The importance of developing a robust synaptic plasticity model and dynamics using distinct levels of abstraction. While this might still be far-fetched, the authors lay out a framework at which these questions need to be posed.
Together, this review paper is timely, well-written, critical, and visionary – an excellent fit for publishing in the IJMS.
Minor comments:
- The authors could discuss the information available on the energy balance at the synapse. A great effort has been invested in quantifying ATP levels both at the presynapse (Tim Ryan’s and Tom Schwartz’s labs, for instance) and postsynapse (Erin Schumann’s work). This might be interesting in the context of phase separation where ATP, at high levels, acts as a hydrotrope (Tony Hyman’s work) regulating the extent of condensation (therefore diffusion, too).
- In Fig. 6D comparing diffusion coefficients with buffering ratios is inadequate. Buffering ratios relate to partitioning coefficients. The authors should replot the data and discuss how diffusion coefficients from FRAP analyses relate to the partitioning of molecules in/out of the SV clusters.
Author Response
Reviewer 1
Reshetniak and colleagues review the state of quantitative approaches to studying synapses, and the potential for marrying with modelling approaches to advance our understanding of synapse biology. The manuscript was a pleasure to read. The discussion of caveats to the imaging data that serves as the basis for quantitative measures as well as the discussion of the relevance of different degrees of abstraction in the modelling was nice. I have only minor comments, and look forward to seeing the paper in print.
We thank the reviewer for the comments.
Minor comments:
L276-286: may be more at home in the antibodies section beginning line 287?
The reviewer is right. We have therefore moved this entire part into the “Antibodies” section. This was effected by simply moving the title “Antibodies” to the original line 289 (now 343).
L314-315: may be useful to clarify that the synaptophysin staining is regular antibody staining.
We have noted this on lines 394-395: “a conventional antibody immunostaining”.
L353-356: the phrasing makes it sound like the cryoelectron tomography studies cited were the first to discover the filamentous network linking synaptic vesicles, but it was well known before (e.g. Landis et al., 1988, Neuron 1(3): 201-209).
We have now cited the respective works. We now write, on lines 445-446: “This work built on previous studies performed using quick freezing, followed by freeze-fracture, etching, and finally rotary angle shadowing (Landis et al., 1988; Hirokawa et al., 1989).”
L382: the section title promises a discussion of potential differences between synapses from different regions and organisms, and how this may impact whether parameters from one model could be transferred to another. I was looking forwards to the discussion about synapses in different organisms, but it wasn’t there.
We have now added a paragraph on this issue, lines 486-501.
L435: Held should be capitalized
We have now corrected this mistake (“Held”).
L467: a reference is made back to “the processes depicted in Fig. 2A”. My memory is not so great, so I had to look back to the figure to find out what this was referencing. It may be appropriate to include a short reminder to help the reader, like “the analysis of selected aspects of clathrin-dependent and -independent vesicle endocytosis depicted in Fig. 2A”, or such sort
We have written “clathrin-dependent and -independent vesicle endocytosis” on the respective line.
L470: “have been develop for”
We have now corrected this mistake (“developed”).
Sudhof’s synaptic vesicle cycle review has a duplicate citation ([43] and [87])
We have now corrected the references.
Reviewer 2 Report
Reshetniak and colleagues review the state of quantitative approaches to studying synapses, and the potential for marrying with modelling approaches to advance our understanding of synapse biology. The manuscript was a pleasure to read. The discussion of caveats to the imaging data that serves as the basis for quantitative measures as well as the discussion of the relevance of different degrees of abstraction in the modelling was nice. I have only minor comments, and look forward to seeing the paper in print.
Minor comments:
L276-286: may be more at home in the antibodies section beginning line 287?
L314-315: may be useful to clarify that the synaptophysin staining is regular antibody staining.
L353-356: the phrasing makes it sound like the cryoelectron tomography studies cited were the first to discover the filamentous network linking synaptic vesicles, but it was well known before (e.g. Landis et al., 1988, Neuron 1(3): 201-209).
L382: the section title promises a discussion of potential differences between synapses from different regions and organisms, and how this may impact whether parameters from one model could be transferred to another. I was looking forwards to the discussion about synapses in different organisms, but it wasn’t there.
L435: Held should be capitalized
L467: a reference is made back to “the processes depicted in Fig. 2A”. My memory is not so great, so I had to look back to the figure to find out what this was referencing. It may be appropriate to include a short reminder to help the reader, like “the analysis of selected aspects of clathrin-dependent and -independent vesicle endocytosis depicted in Fig. 2A”, or such sort
L470: “have been develop for”
Sudhof’s synaptic vesicle cycle review has a duplicate citation ([43] and [87])
Author Response
Reviewer 2
The present review from Sofiia Reshetniak and colleges describes outcomes and suggestions from quantitative approaches to synaptic proteins. This is interesting and fresh work that brings new understanding how the quantitative synaptic data can be combined to construction our view of inside synapse.
I have only minor suggestions for authors.
We thank the reviewer for the comments.
Synapses are characterized by a great variety in sizes, so I recommend to indicate types of synapses which were used in the mentioned investigations counting a number of protein copies.
We now mention this (lines 95, 129, 172, 217-218).
A main principle of lateral membrane organization is lipid raft formation and dynamics. These domains containing high level of cholesterol have a prominent relevance for synaptic organization and transmission (Please see review: PMID: 26332795, 30823359 and ect). I recommend to briefly mention this in the introduction.
We have now included this point (lines 49-50).
In part 2 (lines 66-67) authors could briefly discussed that SV proteins have essential presynaptic functions being inserted in the plasma membrane. For example, after SV exocytosis activity of H-pump can lead to transient acidification of cytoplasm which is important for multiple presynaptic events, especially endocytosis and non-vesicular neurotransmitter release (PMID: 21172612, 21557989).
We have now added this concept (lines 69-73).
Line 85: I suggest to add information related to temperature-dependence of endocytosis (PMID: 27146271; 25296249)
We have now stated this (lines 88-89).
Line 91: Please could authors specify the range of SV population in different synapses.
We have now added a section on the issue of differences between different synapses (lines 486-501).
Line 100: I suggest to note about clathrin- and dynamin independent modes of endocytosis, that can simultaneously operate in synapses, overcoming the limitations of clathrin-driven routes.
We have now noted this on lines 120-121.
In section 3. I think that representation of results showing possible importance of lipid ordering membrane in control of synaptic protein motion and endocytosis looks appropriately.
We now mention this on lines 338-341.
Abstract:
Line 18: Please add “in countless electrophysiological, …. studies.”
We added this word on line 18.
Line 18. I suggest - “The functionality and plasticity…”
We have changed “function” to “functionality” on line 19.
Introduction:
Line 31: Please omit “function”
We deleted “function” on line 31.
Line 32: I suggest:” aberrant synaptic transmission”
We have changed “function” to “transmission” on line 32.
Reviewer 3 Report
The present review from Sofiia Reshetniak and colleges describes outcomes and suggestions from quantitative approaches to synaptic proteins. This is interesting and fresh work that brings new understanding how the quantitative synaptic data can be combined to construction our view of inside synapse.
I have only minor suggestions for authors.
Synapses are characterized by a great variety in sizes, so I recommend to indicate types of synapses which were used in the mentioned investigations counting a number of protein copies.
A main principle of lateral membrane organization is lipid raft formation and dynamics. These domains containing high level of cholesterol have a prominent relevance for synaptic organization and transmission (Please see review: PMID: 26332795, 30823359 and ect). I recommend to briefly mention this in the introduction.
In part 2 (lines 66-67) authors could briefly discussed that SV proteins have essential presynaptic functions being inserted in the plasma membrane. For example, after SV exocytosis activity of H-pump can lead to transient acidification of cytoplasm which is important for multiple presynaptic events, especially endocytosis and non-vesicular neurotransmitter release (PMID: 21172612, 21557989).
Line 85: I suggest to add information related to temperature-dependence of endocytosis (PMID: 27146271; 25296249)
Line 91: Please could authors specify the range of SV population in different synapses.
Line 100: I suggest to note about clathrin- and dynamin independent modes of endocytosis, that can simultaneously operate in synapses, overcoming the limitations of clathrin-driven routes.
In section 3. I think that representation of results showing possible importance of lipid ordering membrane in control of synaptic protein motion and endocytosis looks appropriately.
Abstract:
Line 18: Please add “in countless electrophysiological, …. studies.”
Line 18. I suggest - “The functionality and plasticity…”
Introduction:
Line 31: Please omit “function”
Line 32: I suggest:” aberrant synaptic transmission”
Author Response
Reviewer 3
In this review paper, Reshetniak and colleagues provide an insightful and critical overview of the knowledge and challenges in the quantitative description of synaptic biology. In recent decades, an impressive effort from many labs aimed to identify and quantify the key molecular players at both pre- and post- synapse. Although the repertoire of molecules is well-established, the key concepts that underlie synaptic organization and function are still elusive. In this review, the authors provide a critical overview of the challenges in the field, particularly focusing on:
(i) What the rate-limiting steps are in synaptic processes (i.e., acidification of SVs, clathrin-coat assembly/disassembly, receptor delivery to the postsynapse).
(ii) Technical challenges that arise during data acquisition and interpretation (e.g., variability due to sample preparation, limits of microscopy techniques, labeling, processing extensive data).
(iii) The mobility of proteins sequestered by SVs as a general feature conserved among different types of synapses, further suggesting a common mechanism behind this dynamic sequestering/buffering.
(iv) The importance of developing a robust synaptic plasticity model and dynamics using distinct levels of abstraction. While this might still be far-fetched, the authors lay out a framework at which these questions need to be posed.
Together, this review paper is timely, well-written, critical, and visionary – an excellent fit for publishing in the IJMS.
We thank the reviewer for the comments.
Minor comments:
The authors could discuss the information available on the energy balance at the synapse. A great effort has been invested in quantifying ATP levels both at the presynapse (Tim Ryan’s and Tom Schwartz’s labs, for instance) and postsynapse (Erin Schumann’s work). This might be interesting in the context of phase separation where ATP, at high levels, acts as a hydrotrope (Tony Hyman’s work) regulating the extent of condensation (therefore diffusion, too).
We feel that this discussion would be far beyond the purpose of our review. We focused mainly on protein position, movement and organization, and we would not like to include too many non-related aspects, such as metabolism or ATP concentrations. This would enlarge the review too much, without adding sufficiently to the main purposes of our work.
In Fig. 6D comparing diffusion coefficients with buffering ratios is inadequate. Buffering ratios relate to partitioning coefficients. The authors should replot the data and discuss how diffusion coefficients from FRAP analyses relate to the partitioning of molecules in/out of the SV clusters
The diffusion coefficients from the FRAP analysis relate extremely closely to the partitioning coefficients (R2=0.79, p<0.001). Therefore, we are confident that the diffusion coefficients are also representative for the partitioning coefficients in this context.
The partitioning coefficients were not reported in our original publication (Reshetniak et al., 2020), so it is difficult to plot them here. We therefore prefer to restrict the plot in this review to the published diffusion coefficients, as in our original version, especially as the diffusion coefficients correlate very well with the partitioning coefficients, as mentioned above.